# A genetic model of ivabradine recapitulates results from randomized clinical trials

**Marc-André Legault**[1,2,3], **Johanna Sandoval**[1,3], **Sylvie Provost**[1,3], **Amina Barhdadi**[1,3], **Louis-Philippe Lemieux Perreault**[1,3], **Sonia Shah**[4,5], **R. Thomas Lumbers**[6,7,8], **Simon de Denus**[1,9], **Benoit Tyl**[10], **Jean-Claude Tardif**[1,11]*, **Marie-Pierre Dubé**[1,3,11]*

**1** Montreal Heart Institute, Montreal, Canada, **2** Department of biochemistry and molecular medicine, Université de Montréal, Montreal, Canada, **3** Université de Montréal Beaulieu-Saucier Pharmacogenomics Centre, Montreal, Canada, **4** Institute for Molecular Bioscience, The University of Queensland, Brisbane, Queensland, Australia, **5** Institute of Cardiovascular Science, University College London, London, United Kingdom, **6** Institute of Health Informatics, University College London, London, United Kingdom, **7** Health Data Research UK London, University College London, London, United Kingdom, **8** Bart's Heart Centre, St. Bartholomew's Hospital, London, United Kingdom, **9** Faculty of Pharmacy, Université de Montréal, Montreal, Canada, **10** Cardiovascular Center for Therapeutic Innovation, Institut de Recherches Internationales Servier, Suresnes, France, **11** Department of medicine, Université de Montréal, Montreal, Canada

* marie-pierre.dube@umontreal.ca (MPD); jean-claude.tardif@icm-mhi.org (JCT)

**Data Availability Statement:** Individual level data from the UK Biobank is available to health researchers. The guidelines for the application process are detailed here: https://www.ukbiobank.

## Abstract

### Background

Naturally occurring human genetic variants provide a valuable tool to identify drug targets and guide drug prioritization and clinical trial design. Ivabradine is a heart rate lowering drug with protective effects on heart failure despite increasing the risk of atrial fibrillation. In patients with coronary artery disease without heart failure, the drug does not protect against major cardiovascular adverse events prompting questions about the ability of genetics to have predicted those effects. This study evaluates the effect of a variant in *HCN4*, ivabradine's drug target, on safety and efficacy endpoints.

### Methods

We used genetic association testing and Mendelian randomization to predict the effect of ivabradine and heart rate lowering on cardiovascular outcomes.

### Results

Using data from the UK Biobank and large GWAS consortia, we evaluated the effect of a heart rate-reducing genetic variant at the *HCN4* locus encoding ivabradine's drug target. These genetic association analyses showed increases in risk for atrial fibrillation (OR 1.09, 95% CI: 1.06–1.13, P = 9.3 ×10$^{-9}$) in the UK Biobank. In a cause-specific competing risk model to account for the increased risk of atrial fibrillation, the *HCN4* variant reduced incident heart failure in participants that did not develop atrial fibrillation (HR 0.90, 95% CI: 0.83–0.98, P = 0.013). In contrast, the same heart rate reducing *HCN4* variant did not prevent a composite endpoint of myocardial infarction or cardiovascular death (OR 0.99, 95% CI: 0.93–1.04, P = 0.61).

ac.uk/register-apply/. Selected genetic variants for the construction of genetic scores are available in the Supporting Information files and the software used to compute the scores is available at https://github.com/legaultmarc/grstools. GWAS summary statistics used in the manuscript are publicly available and can be found in the original publications.

**Funding:** This work was supported by the Health Collaboration Acceleration Fund from the Ministère de l'Économie et de l'Innovation du Gouvernement du Québec. M.-A.L. is supported by a Frederick Banting and Charles Best Canada Graduate Scholarship Doctoral Award from the Canadian Institutes of Health Research (CIHR). J.-C.T. holds the Canada Research Chair in personalized medicine and the Université de Montréal Pfizer-endowed research chair in atherosclerosis. M.-P.D. holds the Canada Research Chair in precision medicine data analysis. S.d.D. holds the Université de Montréal Beaulieu-saucier Chair in Pharmacogenomics. SS is partly supported by a National Health and Medical Research Council (NHMRC) fellowship and NHMRC Program Grant 1113400. R.T.L. is supported by a UK Research and Innovation Rutherford Fellowship. Laboratoire Servier provided support in the form of salaries for author B. Tyl but did not have any additional role in the study design, data collection and analysis, decision to publish, or preparation of the manuscript. The specific roles of this author are articulated in the 'author contributions' section.

**Competing interests:** I have read the journal's policy and the authors of this manuscript have the following competing interests: A patent pertaining to pharmacogenomics-guided CETP inhibition (US20190070178A1) owned by Dalcor was granted and J.-C. Tardif and M.-P. Dubé are mentioned as authors but do not receive any royalties. J.-C. Tardif and M.-P. Dubé have a minor equity interest in DalCor. J.-C. Tardif has received research support from Amarin, AstraZeneca, DalCor, Eli-Lilly, Ionis, Pfizer, RegenexBio, Sanofi and Servier, and honoraria from DalCor, Pfizer, Sanofi and Servier. M.-P. Dubé has received honoraria from Dalcor and research support (access to samples and data) from AstraZeneca, Pfizer, Servier, Sanofi and GlaxoSmithKline. B. Tyl is an employee of Laboratoires Servier. Simon de Denus has received grants from Pfizer, AstraZeneca, Roche Molecular Science, DalCor and Novartis. R. Thomas Lumbers has received research grants from Pfizer. The remaining authors have nothing to disclose. This does not alter our adherence to PLOS ONE policies on sharing data and materials.

## Conclusion

Genetic modelling of ivabradine recapitulates its benefits in heart failure, promotion of atrial fibrillation, and neutral effect on myocardial infarction.

## Introduction

Human genetics can be a powerful tool to guide drug development. The identification of mutations in important coronary artery disease associated genes has led to the development of new drugs and the approach of Mendelian randomization (MR) is widely used to predict the effect of interventions on biomarkers [1], to validate drug targets and to predict the effect of drug combinations [2]. There are limitations, however, in the value of human genetics to predict the effects of drugs. The main problems are caused by pleiotropic effects of genetic variants [3], the difference between a lifelong exposure to a risk factor and interventions that are administered after disease onset [2] and the generalizability of the results to specific patient populations and to different ethnic populations [4].

Here, we investigate whether human genetics can reproduce the diverging results obtained on different clinical outcomes in randomized clinical trials of ivabradine. This heart-rate lowering drug was demonstrated to reduce the composite of cardiovascular death and hospitalization for worsening heart failure in patients with symptomatic heart failure and a heart rate above 70 bpm (beats per minute) at baseline in the SHIFT trial [5]. In this study, there was a placebo-adjusted reduction in heart rate of 10.9 (10.4, 11.4) bpm after 28 days on treatment with ivabradine and the hazard ratio (HR) for the cardiovascular composite endpoint was 0.82 (95% CI: 0.75–0.90, p<0.0001). In contrast, in the SIGNIFY trial, ivabradine did not reduce the composite of cardiovascular death or myocardial infarction in patients with stable coronary artery disease (CAD) without heart failure and with a heart rate > 70 bpm at baseline (HR 1.08, 95% confidence interval (CI): 0.96–1.20, P = 0.20) [6]. In both studies, there was an increase in the risk of atrial fibrillation in patients randomized to ivabradine. The incidence of atrial fibrillation was 9% and 8% in the ivabradine and placebo arms in SHIFT (P = 0.012) respectively, whereas it was 5.3% and 3.8% in SIGNIFY (S1 Table). The heart rate reduction induced by ivabradine is due to the inhibition of the "funny" current ($I_f$), which is important for cardiac depolarization during phase 4 of the action potential in the sino-atrial node [7]. The hyperpolarization-activated cyclic nucleotide-gated channel 4 encoded by the *HCN4* gene is responsible for this current [8]. Here, we study naturally occurring variants at this locus as a genetic model of ivabradine therapy to predict the effects of the drug on heart failure, atrial fibrillation and CAD.

## Methods

### Data sources

The UK Biobank is a prospective population cohort of over 500,000 individuals aged between 40 and 69 at recruitment, and has been previously described [9]. We used hospitalization data between the beginning of the Health Episode Statistics (HES) linkage (April 1st 1997) and the last available date for the current data release (March 1st 2016). Codes used to define the clinical variables are presented in S2 Table. All UK Biobank participants were previously genotyped. We applied genetic quality control leaving 413,083 individuals for analysis (Methods in S1 Appendix). All reported genomic positions are reported with respect to build GRCh37. We also used the largest available meta-analysis of genome-wide association studies (GWAS) with

**Abbreviations:** BPM, Beats per minute; MR, Mendelian Randomization; SHIFT, Systolic Heart failure treatment with the $I_f$ inhibitor ivabradine Trial; SIGNIFY, Study Assessing the Morbidity– Mortality Benefits of the $I_f$ Inhibitor Ivabradine in Patients with Coronary Artery Disease; OR, Odds Ratio; HR, Hazard Ratio; CI, Confidence interval; CAD, Coronary Artery Disease; GWAS, Genome- Wide Association Study; GRS, Genetic Risk Score.

summary statistics reporting the effect of the *HCN4* rs8038766 variant on stroke, atrial fibrillation, CAD, myocardial infarction and heart failure (Methods in S1 Appendix) [10–14]. All participants of the UK Biobank gave their informed consent and the present study was approved by the institutional ethics review board of the Montreal Heart Institute.

## Statistical analyses

To identify independent variants at the *HCN4* locus (chr15:73,612,200–73,661,605 ± 200kb) associated with resting heart rate at baseline in the UK Biobank dataset, we used forward stepwise linear regression with additive allele coding and a genome-wide significance threshold ($p \leq 5.0 \times 10^{-8}$). Association between *HCN4* variant rs8038766 and clinical endpoints was assessed using multivariable logistic regression. All models were adjusted for age, sex and the first 10 principal components. For the prospective and competing risk analyses we used Cox proportional-hazards regression. For the competing risk analyses, we estimated the cause-specific hazards where individuals are censored at the time of occurrence of the competing risk if it occurred prior or if it was reported at the same time as the event of interest [15]. We used time from the first baseline visit in years as the timescale and the censure was the date of death or end of follow-up period. For the construction of the heart rate Genetic Risk Score (GRS), we used 64 previously reported genome-wide significant heart rate associated SNPs (with $r^2 <$ 0.1) [16]. We split the participants based on the GRS quintiles with the group formed by the $5^{th}$ GRS quintile (and above) corresponding to the higher heart rate group and the odds ratio for CAD, heart failure and atrial fibrillation were obtained by comparing the first 4 groups individually to the $5^{th}$ group used as reference in logistic regression. All analyses were performed using the R (v.3.5.2) programming language unless otherwise specified.

## Mendelian randomization

We used the inverse variance weighted (IVW) [17], MR-Egger [18], contamination mixture [19] and Mendelian Randomization Pleiotropy RESidual Sum and Outlier (MR-PRESSO) methods [20]. We present results of all four methods in order to outweigh the drawbacks of individual approaches and help guide conclusions, and we report all causal effect estimates for a standard deviation reduction in heart rate as measured in the UK Biobank (11.1 bpm). Coincidentally, this heart rate reduction is similar to the effect of ivabradine in randomized clinical trials (*e.g.* 10.9 bpm in the SHIFT trial) and is comparable in magnitude to the heart rate reduction by ivabradine. For the heart rate GRS, we used the two-stage method with individual level data and the effect estimates are for a 11.1 bpm reduction in heart rate as well [21]. Analyses were performed with the "MendelianRandomization" R package and MR-PRESSO [20]. Refer to Methods in S1 Appendix for additional details.

## Results

### Genetic model of ivabradine

To construct a genetic model of ivabradine treatment, we tested the association between variants at the *HCN4* locus (defined as the gene boundaries ± 200 kb) and heart rate in the UK Biobank by stepwise forward regression analysis. Two independent signals were identified, the first was led by rs8038766 with every copy of the "G" allele reducing heart rate by 0.57 bpm (95% CI 0.51–0.64, P = $2.76 \times 10^{-66}$). This variant is in high linkage disequilibrium (LD) with variants previously associated with resting heart rate, heart rate variability traits and atrial fibrillation (S3 Table). The second variant was rs3743496 with the "T" allele reducing heart rate by 0.30 bpm (95% CI 0.25–0.35, P = $3.96 \times 10^{-30}$). The region spanned by the association

signal led by this variant was wide and overlapped more of the neighbouring gene (*NEO1*), than *HCN4* (S1 Fig). Furthermore, the lead variant was in linkage equilibrium with rs8038766 (D' = 0.43 and $\chi^2$ test of independence p-value <0.0001 in 1000 Genomes phase III Europeans) suggesting that the secondary association signal could in fact not be independent of the first. For these reasons, we were not confident that rs3743496 could be used as a specific and independent genetic instrument of *HCN4* activity and excluded it from the genetic model of ivabradine. Additionally, *HCN4* is a short gene (49,405 bases) that is intolerant to loss-of-function mutations (probability of being loss of function intolerant of 1 in the gnomAD database) [22] which could explain the scarcity of functional variants to be used as genetic instruments.

## Genetically predicted effect of ivabradine on safety endpoints

We tested the association between the heart rate reducing allele of the *HCN4* variant rs8038766 and atrial fibrillation and stroke, a common and well-known consequence of atrial fibrillation. *HCN4* has previously been implicated in atrial fibrillation, and we replicated these results using rs8038766 [23]. In the UK Biobank, rs8038766 was strongly associated with atrial fibrillation (OR 1.09, 95% CI 1.06–1.13; P = $9.3 \times 10^{-9}$) but not with any stroke or ischemic stroke (Fig 1). The association between rs8038766 and atrial fibrillation was also observed in summary statistics from previously published GWAS of atrial fibrillation with OR = 1.11 (P = $1.8 \times 10^{-26}$), and 1.12 (P = $5.4 \times 10^{-35}$) for Roselli *et al.* and Nielsen *et al.* respectively (Fig 1) [12, 13]. Previous epidemiologic studies have shown that chronic atrial fibrillation leads to a five-fold increase in the risk of stroke [24]. We did not find a significant association between rs8038766 and stroke in the UK Biobank, potentially because of the low number of cases (4,158 cases for ischemic stroke). Summary results from the MEGASTROKE consortium show an association between rs8038766 and cardioembolic (OR = 1.08, 95% CI 1.03–1.13, P = $1.54 \times 10^{-3}$) and ischemic stroke (OR = 1.03, 95% CI 1.01–1.05, P = 0.0152) (Fig 1) [25].

## Genetically predicted effect of ivabradine on efficacy endpoints

We tested for association of the heart rate-reducing allele at the *HCN4* variant rs8038766 with combined prevalent and incident heart failure in the UK Biobank and found no association in

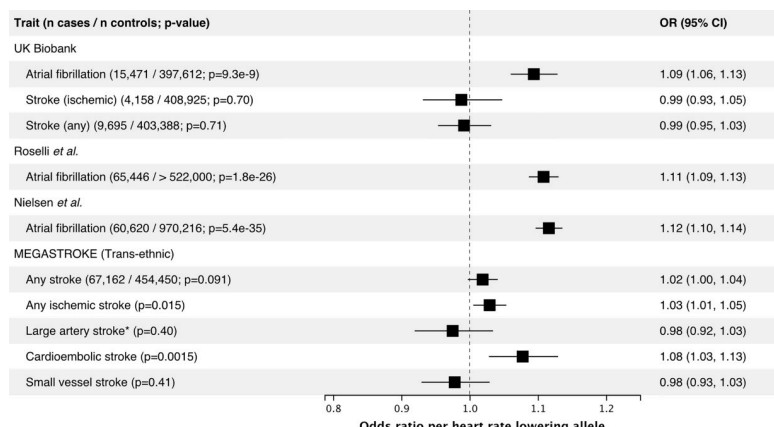

**Fig 1. Association between the heart rate lowering allele (G) of the *HCN4* variant rs8038766 and safety outcomes in the UK Biobank and in published GWAS summary statistics from large consortia.** For the UK Biobank, reporting results from logistic regression comparing the combined prevalent and incident cases to non-cases. References: Roselli et al. [13], Nielsen et al. [12], MEGASTROKE [25]. * rs7174098 (LD $r^2$ = 1 in 1000 genomes Europeans) was used instead of rs8038766 as the latter was unavailable in the MEGASTROKE summary statistics for this outcome.

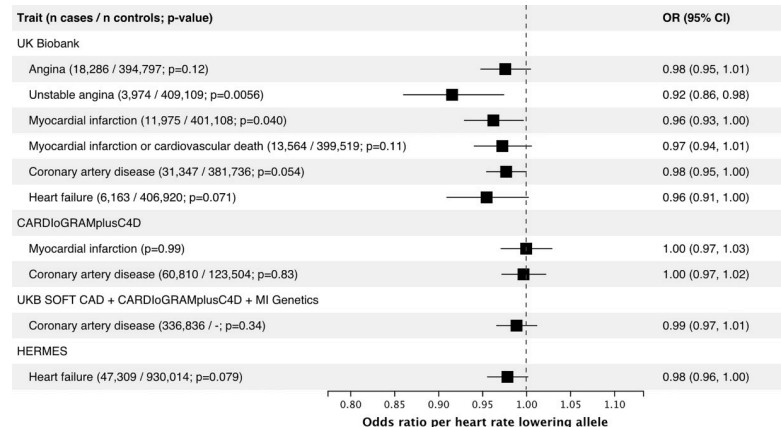

**Fig 2. Association between the heart rate lowering allele (G) of the HCN4 variant rs8038766 and efficacy outcomes in the UK Biobank and in GWAS summary statistics from large consortia.** For the UK Biobank, reporting results from logistic regression comparing the combined prevalent and incident cases to non-cases. References: CARDIoGRAMplusC4D [10], UKB SOFT CAD + CARDIoGRAMplusC4D + MI Genetics [11], HERMES [14].

this naïve model (OR = 0.96, 95% CI 0.91–1.00, p = 0.071) (Fig 2). However, because atrial fibrillation is an important risk factor for heart failure [26], it is a possible that the increased risk of atrial fibrillation attenuates a possible protective association with heart failure. Indeed, after adjustment for any prevalent or incident atrial fibrillation, the association of rs8038766 with heart failure was OR = 0.91, 95% CI 0.87–0.96 (P = $6.7 \times 10^{-4}$). In a model including the interaction term between rs8038766 and atrial fibrillation, the estimated coefficient for the variant was -0.136, 95% CI -0.205, -0.067, (P = 0.00011) and the interaction term coefficient was 0.110 95% CI 0.005, 0.215 (P = 0.04). These coefficients correspond to an estimated OR of the SNP on heart failure of 0.87 in individuals without atrial fibrillation and 0.97 in individuals with atrial fibrillation. However, these associations could be biased if both the SNP and the outcome increase atrial fibrillation risk resulting in a possible collider bias. To account for this, we used a cause-specific hazards model for the incidence of heart failure and atrial fibrillation separately using 404,767 UK Biobank participants that were free of both diseases at baseline. In this group, there were 3,385 incident heart failure cases and the *HCN4* variant rs8038766 showed a non-significant trend for a protective effect (HR = 0.96, 95% CI: 0.89–1.02; P = 0.177) (Table 1). However, in a competing risk model accounting for incident occurrences of atrial fibrillation, the protective effect of the heart rate-reducing variant on heart failure was brought to focus with HR = 0.90, 95% CI 0.83–0.98 (P = 0.013) (Table 1). We conducted a similar analysis using incident myocardial infarction or cardiovascular death corresponding to the primary endpoint in the SIGNIFY trial, which was also potentially exposed to the opposing effects of the heart rate-reducing variant on atrial fibrillation and myocardial infarction. There was no detectable association of the heart rate-reducing variant with myocardial infarction or cardiovascular death in the simple Cox proportional-hazards model (HR = 0.99, 95% CI 0.94–1.05) or in the cause-specific competing risk model (HR = 0.99, 95% CI 0.93–1.04) (Table 1). We did see, however, an association between rs8038766 and prevalent or incident cases of unstable angina (OR 0.92 95% CI 0.86–0.98, P = 0.0056) (Fig 2).

In the HERMES case-control consortium, the heart rate reducing allele of rs8038766 was only weakly associated with heart failure (OR 0.98 95% CI 0.96–1.00, p = 0.079) (Fig 2), but when using the mtCOJO method to adjust for atrial fibrillation using summary statistics [27], the protective effect was increased with a conditional OR = 0.96 95% CI 0.94–0.98

**Table 1. Association of the *HCN4* variant with outcomes in the UK Biobank using prospective and cause-specific hazard competing risk analyses.**

| Outcome | Model | N total | N events | HR (95% CI) * | P-value |
|---|---|---|---|---|---|
| Genetic model for SHIFT** | | | | | |
| *Using participants without a history of atrial fibrillation or heart failure at recruitment* | | | | | |
| **Heart failure** | Cox proportional-hazards | 404,767 | 3,385 | 0.96 (0.89, 1.02) | 0.18 |
| **Atrial fibrillation** | Cox proportional-hazards | 404,767 | 8,461 | 1.08 (1.04, 1.13) | $9.4 \times 10^{-5}$ |
| **Heart failure** | Competing risk (atrial fibrillation) | 404,767 | 2,380 | 0.90 (0.83, 0.98) | 0.013 |
| **Atrial fibrillation** | Competing risk (heart failure) | 404,767 | 7,663 | 1.08 (1.04, 1.13) | $3.2 \times 10^{-4}$ |
| Genetic model for SIGNIFY** | | | | | |
| *Using participants without a history of atrial fibrillation or MI at recruitment* | | | | | |
| **MI or CV Death** | Cox proportional-hazards | 397,008 | 4,976 | 0.99 (0.94, 1.05) | 0.84 |
| **Atrial fibrillation** | Cox proportional-hazards | 397,008 | 7,880 | 1.08 (1.04, 1.13) | $3.1 \times 10^{-4}$ |
| **MI or CV Death** | Competing risk (atrial fibrillation) | 397,008 | 4,534 | 0.99 (0.93, 1.04) | 0.61 |
| **Atrial fibrillation** | Competing risk (MI or CV death) | 397,008 | 7,482 | 1.09 (1.04, 1.13) | $1.4 \times 10^{-4}$ |

* Reporting the effect of the heart rate reducing allele of rs8038766 at the *HCN4* gene. All models were adjusted for age, sex and the first 10 principal components. In the Cox proportional-hazards models, individuals were censored at the time of death or end of follow up; in the competing risk models, individuals were censored at the time of occurrence of the competing event, death, or end of follow up.

** Our model aims to match the outcome and exposure of interest from the SHIFT and SIGNIFY trials, but we did not emulate the trials in any other way such as by matching the inclusion / exclusion criteria.

CV, cardiovascular; HR, hazard ratio; MI, myocardial infarction.

(P = $9.7 \times 10^{-4}$) [14]. In the CARDIoGRAMplusC4D consortium, there was no association between rs8038766 and CAD or myocardial infarction (Fig 2).

## Bi-directional MR

Bi-directional MR supports a causal effect of atrial fibrillation on heart failure with a causal OR estimate of 1.22 and ranging up to 1.25 according to different MR models (Table 2, S7 Table), and supports a causal effect of heart failure on atrial fibrillation with OR ranging from 1.21–1.94 (excluding the contamination mixture model estimate of 6.82 which is an outlier among the other methods). These results are concordant with observational longitudinal studies that have observed an increased incidence of atrial fibrillation in new heart failure patients and vice versa and where both diseases are often diagnosed on the same day [26]. Bi-directional MR

**Table 2. Bi-directional Mendelian randomization estimates.**

| Exposure | Outcome | MR Causal OR (95% CI) * | P-value |
|---|---|---|---|
| Atrial fibrillation (152 variants) | Heart failure | 1.23 (1.20, 1.27) | $3.7 \times 10^{-52}$ |
| Atrial fibrillation (152 variants) | Coronary artery disease | 1.00 (0.98, 1.03) | 0.76 |
| Atrial fibrillation (152 variants) | Myocardial infarction | 0.98 (0.95, 1.02) | 0.30 |
| Heart failure (11 variants) | Atrial Fibrillation | 1.45 (1.11, 1.90) | 0.0067 |
| Coronary artery disease (68 variants) | Atrial Fibrillation | 1.15 (1.11, 1.21) | $1.7 \times 10^{-10}$ |
| Myocardial infarction (31 variants) | Atrial Fibrillation | 1.11 (1.06, 1.16) | $1.3 \times 10^{-5}$ |

Summary statistics for atrial fibrillation taken from Nielsen et al. [12] for myocardial infarction and CAD from CARDIoGRAMplusC4D and CARDIoGRAMplusC4D + UKB SOFT + MiGen [10, 11], for heart failure from HERMES [14].

* IVW MR model. For MR results using MR-Egger, the contamination mixture model and MR-PRESSO, see S7 Table. Causal ORs relate the odds of the outcome in exposed individuals vs non-exposed.

IVW, inverse-variance weighted; MR, Mendelian randomization; OR, odds ratio.

also supported a causal effect of CAD and myocardial infarction on atrial fibrillation, but not the opposite. The point estimates ranged from OR: 1.12–1.17 for the effect of CAD on atrial fibrillation and OR: 1.11–1.22 for the effect of myocardial infarction on atrial fibrillation (Table 2, S7 Table).

### Effect of heart rate on cardiovascular outcomes

To compare our observations of heart rate reduction attributable to the *HCN4* variant to that of polygenic origin, we constructed a genetic risk score (GRS) with 64 variants previously associated with heart rate (S4 Table) [16]. An increase of 1 standard deviation in the heart rate GRS was associated with a 1.76 (1.73, 1.80) bpm increase in heart rate explaining 2.5% of the variance in the UK Biobank data (S2 Fig). The two stage method causal estimate scaled for a 11.1 bpm (corresponding to 1 standard deviation of heart rate in the UK Biobank and concordant with the effect of ivabradine in clinical trials) genetic reduction in heart rate was OR = 1.25 (95% CI: 1.13–1.39) for atrial fibrillation, OR = 1.03 (95% CI: 0.88–1.21) for heart failure and 1.03 (95% CI: 0.96–1.11) for CAD. To account for the presence of pleiotropy, we also used MR-Egger, contamination mixture model and the MR-PRESSO methods (S5 Table), and saw an increase in the risk for atrial fibrillation associated with heart rate reduction (OR 1.54, $P = 1.3 \times 10^{-7}$ for MR-PRESSO), but not with heart failure or CAD, although these did not take into account the possible competing effect of atrial fibrillation. MR analyses using effect estimates derived from larger GWAS consortia using the same set of 64 heart rate variants supported results observed with the UK Biobank data (S6 Table).

## Discussion

In the present study, we used genetics to infer the causal effect of ivabradine on safety and efficacy outcomes in an attempt to reproduce observations from randomized clinical trials, and to assess the value of genetic approaches to support drug targets and trial design issues such as target patient population and clinical outcomes.

### Effect of HCN4 on atrial fibrillation

Genetically predicted heart rate reduction from the *HCN4* gene variant rs8038766 was associated with an increase in risk of atrial fibrillation, recapitulating the observations from the SIGNIFY and SHIFT trials. In our MR analyses using methods robust to the inclusion of invalid instruments, we observed that a genetically predicted reduction in heart rate of approximately 11 bpm conferred an increased risk of atrial fibrillation with an OR of 1.54 in the UK Biobank and OR ranging from 1.36 to 1.56 using summary statistics from previous large GWAS. These results are also coherent with a recent MR study reporting a protective effect of increased heart rate for atrial fibrillation and cardioembolic stroke [28]. Atrial fibrillation is known to increase the risk of stroke by 3 to 5-fold [24]. In clinical trials of ivabradine, there was no treatment association with stroke, but the small number of incident atrial fibrillation events would have made such an observation unlikely [29]. The estimated effects of heart rate on atrial fibrillation are smaller than the effect predicted from the *HCN4* variant alone, whose scaled OR for a comparable 11 bpm reduction would be greater than 5. This may be explained partly by the inaccuracy of extrapolation of the OR estimate derived from a single genetic variant, and also possibly by an effect of *HCN4* on atrial fibrillation that may be specific to modulation of the $I_f$ current or other structural consequences of *HCN4* variants. For example, genetic mutations in *HCN4* have been associated to Brugada syndrome and sick sinus syndrome as well as left ventricular noncompaction and it is possible that common polymorphisms in the *HCN4* gene have more subtle effects on myocardium structure or conduction parameters that may be

independent of heart rate [30, 31]. Additionally, altering *HCN4* function or levels during embryogenesis in other species have been shown to structurally alter heart development which could explain effects beyond heart rate modulation alone [32]. The MR estimates from the UK Biobank are also based on mostly healthy individuals with a low heart rate (mean of 69 bpm) possibly limiting clinical interpretation [33]. However, the effect estimates of heart rate reduction on atrial fibrillation are similar when using the GWAS results from Nielsen *et al.* which include both population-based and clinical cohorts [12]. The possibility that the effect is greater in healthy patients is also supported by the previously described association between low heart rate during physical activity and the increased incidence of atrial fibrillation [34].

## Effect of HCN4 on ischemic endpoints

We tested the association between the *HCN4* heart rate-reducing variant and various ischemic endpoints in the UK Biobank. The largest effect we observed was with unstable angina, which is coherent with the use of ivabradine to alleviate anginal symptoms. Nonetheless, the effect sizes of the association with CAD, angina and myocardial infarction were small and marginally significant in the UK Biobank and importantly they were not supported by results from larger GWAS consortia. This suggests that the effect of *HCN4* on CAD may be null or of a very small effect size so as to not be detectable in the context of a clinical trial such as in the SIGNIFY study [6]. We also investigated the possibility that the increased risk of atrial fibrillation offsets the beneficial effects on the SIGNIFY primary endpoint of myocardial infarction or cardiovascular death using a prospective competing risk analysis in individuals that did not develop atrial fibrillation and showed that accounting for atrial fibrillation had no impact on the risk for myocardial infarction or cardiovascular death. There was no detectable association of the *HCN4* heart rate-reducing variant with myocardial infarction or cardiovascular death in the cause-specific competing risk model. This was further supported by the bi-directional MR analysis that showed that CAD caused atrial fibrillation but not the opposite. Finally, the MR study did not show a causal link between heart rate and CAD suggesting that reducing heart rate is not sufficient to prevent the disease.

## Relationship with clinical trials of ivabradine

The analysis of the subgroup of participants with angina class 2 or greater at baseline in SIG-NIFY showed a nominal increase in the rate of the primary endpoint of cardiovascular death or myocardial infarction with ivabradine [5, 6]. Whether this observation represented a chance finding in the context of a neutral result in the overall SIGNIFY population or a potential signal of harm in this subset of patients was a matter of discussion. The results of the current analyses support neutral effects of $I_f$ current inhibition on the composite endpoint of cardiovascular death and myocardial infarction, without evidence of harm.

In the SHIFT trial, ivabradine reduced the rate of the primary composite endpoint of cardiovascular death or hospitalization for worsening heart failure in patients with heart failure with reduced ejection fraction and without atrial fibrillation. In the genetic model of ivabradine, the competing risk analysis accounting for atrial fibrillation showed that the *HCN4* heart rate-reducing variant protected against heart failure (HR = 0.90, 95% CI: 0.83–0.98, P = 0.013). The results from the marginal models and the competing risk analyses do suggest opposing effects of the heart rate-reducing *HCN4* variant on atrial fibrillation and heart failure. The importance of these effects is also highlighted by the bi-directional MR of atrial fibrillation and heart failure that confirmed that both diseases are mutually causal of one another.

## Study limitations

As for any MR study, our analyses were subject to the assumptions of the underlying models and the possibility of unobserved horizontal pleiotropy. Additionally, our genetic model of ivabradine corresponds to a lifelong effect as opposed to an exposure after drug initiation. Generally, common variants also result in effects of smaller magnitude than ones resulting from pharmacological modulation and extrapolation is required to compare them. We also used data from individuals of predominantly European ancestry both in the UK Biobank and in summary statistics from large GWAS consortia which could limit the generalizability of our results to other populations both in terms of clinical profile and ancestry. Finally, we defined clinical variables based on combinations of hospitalization and death record codes in the UK Biobank which is likely to result in imperfect coding of disease status.

## Conclusion

In conclusion, genetic modelling of ivabradine recapitulates its benefits in heart failure, promotion of atrial fibrillation, and neutral effect on myocardial infarction. This study supports the use of methods that leverage naturally occurring genetic variants to predict diverging results on different clinical outcomes and support the design of randomized clinical trials, even in a situation where more complex disease risks are at play.

## Supporting information

**S1 Appendix. Supplementary methods and references for supplementary methods, figures and tables.**
(DOCX)

**S1 Fig. Results from the stepwise regression of HCN4 variants on heart rate in the UK Biobank.**
(DOCX)

**S2 Fig. Effect of heart rate genetic risk score groups based on quintiles on atrial fibrillation, heart failure and coronary artery disease in the UK biobank dataset.**
(DOCX)

**S1 Table. Summary of ivabradine cardiovascular outcomes trials.**
(DOCX)

**S2 Table. Self-reported, hospitalization (ICD10) and operation (OPCS) codes used to define clinical variables based on the UK Biobank available data.**
(DOCX)

**S3 Table. Results from the NHGRI-EBI GWAS catalog mapped to the *HCN4* gene.**
(DOCX)

**S4 Table. Variants and weights used for the computation of the heart rate GRS.**
(DOCX)

**S5 Table. MR estimates based on 64 heart-rate associated variants and their effect on outcomes in the UK Biobank.**
(DOCX)

**S6 Table. MR estimates based on the effect of 64 heart-rate associated variants in external summary statistics from large GWAS consortia.**
(DOCX)

**S7 Table. Participant overlap between GWAS meta-analysis studies used for observational and Mendelian randomization analyses and the UK Biobank.**
(DOCX)

## Acknowledgments

This research has been conducted using the UK Biobank Resource under Application Number 20168.

## Author Contributions

**Conceptualization:** Marc-André Legault, Simon de Denus, Benoit Tyl, Jean-Claude Tardif, Marie-Pierre Dubé.

**Data curation:** Johanna Sandoval, Sylvie Provost.

**Formal analysis:** Marc-André Legault.

**Funding acquisition:** Marie-Pierre Dubé.

**Investigation:** Marc-André Legault, R. Thomas Lumbers, Benoit Tyl, Jean-Claude Tardif, Marie-Pierre Dubé.

**Methodology:** Marc-André Legault, Sylvie Provost, Amina Barhdadi, Louis-Philippe Lemieux Perreault, Marie-Pierre Dubé.

**Project administration:** Sylvie Provost, Marie-Pierre Dubé.

**Resources:** Sonia Shah, R. Thomas Lumbers.

**Software:** Marc-André Legault, Louis-Philippe Lemieux Perreault.

**Supervision:** Marie-Pierre Dubé.

**Visualization:** Marc-André Legault.

**Writing – original draft:** Marc-André Legault.

**Writing – review & editing:** Marc-André Legault, Jean-Claude Tardif, Marie-Pierre Dubé.

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
