## [Decision Letter · Decision Letter 0]

13 May 2020

PONE-D-20-09954

A genetic model of ivabradine recapitulates results from randomized clinical trials

PLOS ONE

Dear Dr. Dubé,

Thank you for submitting your manuscript to PLOS ONE. After careful consideration, we feel that it has merit but does not fully meet PLOS ONE’s publication criteria as it currently stands. Therefore, we invite you to submit a revised version of the manuscript that addresses the points raised during the review process.

We would appreciate receiving your revised manuscript by Jun 27 2020 11:59PM. To enhance the reproducibility of your results, we recommend that if applicable you deposit your laboratory protocols in protocols.io, where a protocol can be assigned its own identifier (DOI) such that it can be cited independently in the future. For instructions see: http://journals.plos.org/plosone/s/submission-guidelines#loc-laboratory-protocols

We look forward to receiving your revised manuscript.

Kind regards,

Ify Mordi

Academic Editor

PLOS ONE

Journal Requirements:

I have read the journal's policy and the authors of this manuscript have the following competing interests: A patent pertaining to pharmacogenomics-guided CETP inhibition was granted and J.-C.Tardif and M.-P. Dubé are mentioned as authors. J.-C.Tardif and M.-P. Dubé have a minor equity interest in DalCor. J.-C.Tardif has received research support from Amarin, AstraZeneca, DalCor, Eli-Lilly, Ionis, Pfizer, RegenexBio, Sanofi and Servier, and honoraria from DalCor, Pfizer, Sanofi and Servier. M.-P. Dubé has received honoraria from Dalcor and research support (access to samples and data) from AstraZeneca, Pfizer, Servier, Sanofi and GlaxoSmithKline. B. Tyl is an employee of Laboratoires Servier. Simon de Denus has received grants from Pfizer, AstraZeneca, Roche Molecular Science, DalCor and Novartis. R. Thomas Lumbers has received research grants from Pfizer. The remaining authors have nothing to disclose.

We note that one or more of the authors are employed by a commercial company: Laboratoires Servier

We also note that you have a patent relating to material pertinent to this article. Please ensure your amended statement of Competing Interests declares this patent (with details including name and number), along with any other relevant declarations relating to employment, consultancy, patents, products in development or modified products etc.

Within your Competing Interests Statement, please confirm that this commercial affiliation and patent does not alter your adherence to all PLOS ONE policies on sharing data and materials by including the following statement: "This does not alter our adherence to  PLOS ONE policies on sharing data and materials.” (as detailed online in our guide for authors http://journals.plos.org/plosone/s/competing-interests) . If this adherence statement is not accurate and  there are restrictions on sharing of data and/or materials, please state these. Please note that we cannot proceed with consideration of your article until this information has been declared.

Reviewers' comments:

Reviewer's Responses to Questions

**Comments to the Author**

1. Is the manuscript technically sound, and do the data support the conclusions?

Reviewer #1: Yes

Reviewer #2: Yes

Reviewer #3: Yes

2. Has the statistical analysis been performed appropriately and rigorously? 

Reviewer #1: Yes

Reviewer #2: Yes

Reviewer #3: Yes

3. Have the authors made all data underlying the findings in their manuscript fully available?

Reviewer #1: Yes

Reviewer #2: Yes

Reviewer #3: Yes

4. Is the manuscript presented in an intelligible fashion and written in standard English?

Reviewer #1: Yes

Reviewer #2: Yes

Reviewer #3: Yes

5. Review Comments to the Author

Reviewer #1: The authors studied genetic variants in the HCN4 gene for association with heart failure, atrial fibrillation and coronary artery disease. They reconfirm that genetic variants in this region, which primarily associate with heart rate, have also significant association with atrial fibrillation and mildly with cardioembolic stroke. There were also some signals for association with heart failure, particularly after adjustment for atrial fibrillation. Moreover, the authors studied a genetic score for heart rate and cardiovascular outcome and confirm that a reduction in heart rate is positively associated with atrial fibrillation but no other obvious cardiovascular conditions. The manuscript largely recapitulates findings that have been made for genetic variants at the HCN4 locus published in genomewide association studies. The strength of the paper is the focus on predicting Ivabradine related effects (i.e. pharmacological effects similar to those of genetic variants) as observed in previous clinical trials.

Minor comments

It would be interesting to see whether the genetic variants tested affect expression of the HCN4 gene as can be studied in publically available databases.

In the context of the reported variants, the authors should not use the term mutation.

Reviewer #2: Sorry, I don't have a lot of time to re-read. I reviewed this manuscript at a previous journal and liked it, but it wasn't prioritized for publication there. I've copy-pasted my comments from the previous review - if any comments have already been addressed in the editing process between journals, please indicate that the comment is no longer relevant. Thanks and best wishes, Steve

---

Please note that I am a statistician, so my comments relate to the statistical aspects of the analysis - I am not best placed to comment on other aspects.

Generally speaking, this was a persuasive and thorough exposition of the interrelations between various cardiovascular diseases. I have only minor comments:

1) I'd appreciate if someone with more knowledge than me was able to comment on the use of the rs8038766 variant only, and the fact that the rs3743496 variant was ignored. Having a high LD-score isn't necessarily a bad thing (depends on what its neighbours are), and you would expect a variant that is selected in a region-wide search to have a higher LD score than average (and 87th %ile isn't especially high!). The argument that the variant overlaps another gene region is a more persuasive reason for excluding it from the analysis, but many variants have LD regions that overlap different genes. Anyway, I trust that this was a decision made for principled reasons rather than for convenience (the results looked better when looking at one variant only), but the reasons given seem to me a little thin.

2) Estimates in the tables in units. In Table 2, what are the units for the causal ORs? (are they per unit increase in the log odds of the exposure?). In Figures 1/2, could write "Odds ratio per heart rate lowering allele" on the figure (this info is in the legend). Is there any systematic difference between these figures? In the Take-home figure, I'm not fully clear what "Genetic model of SHIFT" means. I presume 11 bpm is 1 SD for heart rate, but it could be the mean effect of taking ivabradine.

3) I'd appreciate if someone with more knowledge than me was able to comment on the plausibility of the bidirectional analyses. Is it plausible to suggest that there could be bidirectional effects of AF on heart failure? Does AF sometimes precede HF, and sometimes HF precedes AF? If not, this doesn't invalidate the analyses (could be that the genetic variants are picking up an effect of subclinical HF on AF risk or vice versa), but it changes the interpretation - in my (limited) experience, true bidirectional effects are rare.

4) This is really picky, but generally the term "2SLS method" implies a continuous outcome and a linear regression model. If you used linear regression and treated the binary outcome as continuous, then fair enough (it doesn't make much difference). But if you used logistic/Cox PH regression, then "two-stage method" is my preferred term (others have been suggested).

Otherwise, I don't have much else to say - it was an interesting read!

---

Stephen Burgess

Reviewer #3: The authors present their findings on using genetic variation in HCN4, the gene which encodes the target of ivabradine, to replicate the findings of previously published drug trials of ivabradine. Major strengths are the various complementary analyses using large-scale data sources and the detailed documentation. Overall the manuscript is well-written. Some observations and questions:

- It seems that the UK Biobank was a large contributor to many of the sources of summary statistics. It would be good to provide the reader insight in the % sample overlap (always with respect to the larger study) for the various data sources which are combined across the different analyses.

- Given that the genetic risk score for heart rate was derived from a GWAS meta-analysis where the UK Biobank formed the discovery stage, any MR analyses performed in the one-sample setting of the UK Biobank with this score might suffer from the Winner’s Curse. How would this influence the reported results?

- On page 10 the authors describe how adjusting for atrial fibrillation would be problematic if both the SNP and heart failure collide on AF, which would introduce collider bias. However, isn’t it equally likely and problematic that, if the SNP has an effect on AF, and AF has an effect on heart failure, that both the SNP and the confounders of the AF-heart failure association would collide on AF (leading to collider bias when you adjust for AF)?

- The interpretation of main effect estimates are less straightforward when interaction effects have been added to the model. Therefore, please be explicit how the reader should interpret the sentence providing both the main and interaction (with AF) estimates between rs8038766 and heart failure.

- Figure 1: Why was rs7174098 used for just one outcome in MEGASTROKE?

- Supposedly you choose the transethnic data of MEGASTROKE for its large number of cases. However, rs8038766 need not necessarily be a strong genetic proxy for HCN4 in non-Europeans. Does using METASTROKE's European dataset show comparable results?

- Please report (perhaps in supplemental material) whether the various GWAS meta-analyses were based on incident or prevalent cases and whether recurrent events were included.

- There exist MR methods which can incorporate correlated genetic variants to boost power. Did you consider these methods for HCN4-variants?

- Table 1 describes the ‘genetic model for SHIFT/SIGNIFY’. Please be explicit this only refers to the outcome definition (and intervention), i.e., not also the inclusion criteria for participants.

- The selected population of the UK Biobank may give rise to issues like selection bias, also for MR studies. Could this have influenced your results?

Minor:

- Please mention that the heart rate GRS includes variants which are just relatively independent (r2<0.1). Lower r2 thresholds are now typically advised for MR (e.g., 0.001). In extension, please be more explicit regarding the independence of the various sets of instruments used in the bidirectional MR analyses.

- Perhaps of interest, FINNGEN could serve as an additional source of publicly available summary statistics for (ICD-code based) heart failure

- For the analyses with the GRS it seems two-sample methodology was applied in the one-sample MR setting (e.g., Supplemental Table 5). Perhaps of interest: this (very) recent preprint (https://www.biorxiv.org/content/10.1101/2020.05.07.082206v1) suggests that particularly the MR-Egger method may be easily biased in this setting. Calculating the I2 may give insight.

- Not a fan of ‘non-significant trend’ – perhaps rephrase to ‘provided (very) weak evidence’?

- With regard to the kinship threshold of >0.0884 – wouldn’t this reflect 2nd degree relationships or more (rather than ‘or less’)?

- Please note that quintiles are not the group themselves but rather the cut-offs to define these groups

- The abbreviation GRS is introduced twice in the methods section

6. PLOS authors have the option to publish the peer review history of their article (what does this mean?). If published, this will include your full peer review and any attached files.

Reviewer #1: No

Reviewer #2: Yes: Stephen Burgess

Reviewer #3: No

---

## [Author Response · Author response to Decision Letter 0]

17 Jun 2020

Editor comment 1: Style requirements

We have now modified the manuscript as needed to comply with the PLOS One style requirements.

Editor comment 2: competing interests and funding statements 

We have updated the competing interests and funding statements as required to highlight the possible role of funders in our study. The new statements were enclosed within the cover letter. Note that the patent on pharmacogenomics-guided CETP inhibition is only distantly related to the topic of the manuscript as it concerns a drug of a different class (CETP-inhibition) and used for a different indication than ivabradine. We will clarify the information in the individual declarations of Competing Interests. 

The patent title is “Methods for Treating or Preventing Cardiovascular Disorders and Lowering Risk of Cardiovascular Events”, the patent number is US20190070178A1, the patent is owned by Dalcor and authors receive no royalties. 

Editor comment 4: Supporting information

We have added captions for the S1 Appendix, S1-S2 Figures and S1-S7 Tables at the end of the manuscript. We also updated the in-text references.

Reviewer’s comments to the Authors :

Reviewer #1: 

The authors studied genetic variants in the HCN4 gene for association with heart failure, atrial fibrillation and coronary artery disease. They reconfirm that genetic variants in this region, which primarily associate with heart rate, have also significant association with atrial fibrillation and mildly with cardioembolic stroke. There were also some signals for association with heart failure, particularly after adjustment for atrial fibrillation. Moreover, the authors studied a genetic score for heart rate and cardiovascular outcome and confirm that a reduction in heart rate is positively associated with atrial fibrillation but no other obvious cardiovascular conditions. The manuscript largely recapitulates findings that have been made for genetic variants at the HCN4 locus published in genomewide association studies. The strength of the paper is the focus on predicting Ivabradine related effects (i.e. pharmacological effects similar to those of genetic variants) as observed in previous clinical trials.

Author response: Thank you for your comments and agreeing to review our manuscript.

Minor comments

Comment #1: It would be interesting to see whether the genetic variants tested affect expression of the HCN4 gene as can be studied in publically available databases.

Author response: Thank you for this suggestion. We agree that using eQTLs of HCN4 would have been ideal, however the tissue of interest in this study is the sinoatrial node which is not available in gene expression resources. As such, the identification of strong eQTLs valuable for MR in our study is not possible.

Comment #2: In the context of the reported variants, the authors should not use the term mutation.

Author response: As suggested by the reviewer, we have adapted the wording where we refer to common HCN4 polymorphisms. Specifically, we replaced “mutation” by “variant” when referring to common variants everywhere in the manuscript.

Reviewer #2: 

Sorry, I don't have a lot of time to re-read. I reviewed this manuscript at a previous journal and liked it, but it wasn't prioritized for publication there. I've copy-pasted my comments from the previous review - if any comments have already been addressed in the editing process between journals, please indicate that the comment is no longer relevant. Thanks and best wishes, Steve

Please note that I am a statistician, so my comments relate to the statistical aspects of the analysis - I am not best placed to comment on other aspects.

Generally speaking, this was a persuasive and thorough exposition of the interrelations between various cardiovascular diseases. I have only minor comments:

Author response: Thank you for your comments and accepting to review our manuscript. We had included many of your previous recommendations into the current version of the manuscript. We will highlight the changes in response to the specific points below.

Comment #1: I'd appreciate if someone with more knowledge than me was able to comment on the use of the rs8038766 variant only, and the fact that the rs3743496 variant was ignored. Having a high LD-score isn't necessarily a bad thing (depends on what its neighbours are), and you would expect a variant that is selected in a region-wide search to have a higher LD score than average (and 87th %ile isn't especially high!). The argument that the variant overlaps another gene region is a more persuasive reason for excluding it from the analysis, but many variants have LD regions that overlap different genes. Anyway, I trust that this was a decision made for principled reasons rather than for convenience (the results looked better when looking at one variant only), but the reasons given seem to me a little thin.

Author response: We agree with the reviewer’s assessment that the selection of a single variant as genetic instrument is not ideal. This is our only course of action, unfortunately, as HCN4 is a short and highly constrained gene with very few variants known to affect function (pLI = 1 on gnomAD v2.1.1).

We have opted to not use the second variant identified in the stepwise analysis, rs3743496, because it spans an association signal that overlaps more with the NEO1 gene than the HCN4 gene (Online Figure 1). Furthermore, the Online Figure 1 suggests residual LD between both signals questioning the true independence of both signals. In the 1000 genomes phase III Europeans, the D’ between both SNPs was 0.43 suggesting a strong deviation from independence even though the r2 was 0.02. This discrepancy between metrics of linkage disequilibrium can be explained by the large difference in MAF (0.16 for rs8038766 and 0.38 for rs3743496). In light of these findings, we believe that the use of rs8038766 alone is likely to be a good proxy for the true HCN4 modulating variants and that the inclusion of the second variant would not lead to added benefit. Moreover, there are more independently identified GWAS associations with heart rate parameters and heart rate variability traits in SNPs in LD with rs8038766 than with rs3743496 (Supplementary Table 3) suggesting that it is a better tag for HCN4 functional variants.

We have adapted the text to add these justifications. The “results / genetic model of ivabradine” section now reads:

“This variant is in high linkage disequilibrium (LD) with variants previously associated with resting heart rate, heart rate variability traits and atrial fibrillation (Supplementary Table 3). The second variant was rs3743496 with the “T” allele reducing heart rate by 0.30 bpm (95% CI 0.25-0.35, P=3.96 × 10-30). The region spanned by the association signal led by this variant was wide and overlapped more of the neighbouring gene (NEO1), than HCN4 (Online Figure 1). Furthermore, the lead variant was in linkage equilibrium with rs8038766 (D’ = 0.43 and �2 test of independence p-value <0.0001 in 1000 Genomes phase III Europeans) suggesting that the secondary association signal could in fact not be independent of the first. For these reasons, we were not confident that rs3743496 could be used as a specific and independent genetic instrument of HCN4 activity and excluded it from the genetic model of ivabradine. Additionally, HCN4 is a short gene (49,405 bases) that is intolerant to loss-of-function mutations (probability of being loss of function intolerant of 1 in the gnomAD database) 22 which could explain the scarcity of functional variants to be used as genetic instruments.”

Comment #2: Estimates in the tables in units. In Table 2, what are the units for the causal ORs? (are they per unit increase in the log odds of the exposure?).

Author response: We have added that the IVW OR estimates from Table 2 are to be interpreted as the increase in odds of the outcome in exposed vs non-exposed.

In Figures 1/2, could write "Odds ratio per heart rate lowering allele" on the figure (this info is in the legend). Is there any systematic difference between these figures?

Author response: We have adapted Figures 1 and 2 to read “Odds ratio per heart rate lowering allele” as recommended.

Figures 1 and 2 are very similar but Figure 1 shows results for safety outcomes whereas Figure 2 shows results for efficacy outcomes (with respect to ivabradine treatment). Apart from that distinction there are no systematic differences.

In the Take-home figure, I'm not fully clear what "Genetic model of SHIFT" means. I presume 11 bpm is 1 SD for heart rate, but it could be the mean effect of taking ivabradine.

Author response: We have removed the “take home” figure for this journal as it was not required. To clarify, the 11 bpm does correspond to 1 s.d. of heart rate in the UK Biobank, but the effect of taking ivabradine in RCTs was comparable in magnitude (e.g. placebo adjusted net reduction of 10.9 bpm in ivabradine arm of the SHIFT trial at 28 days). To clarify this important point, we have added the following text in the “Mendelian randomization” subsection of the “Methods” section:

“We present results of all four methods in order to outweigh the drawbacks of individual approaches and help guide conclusions, and we report all causal effect estimates for a standard deviation reduction in heart rate as measured in the UK Biobank (11.1 bpm). Coincidentally, this heart rate reduction is similar to the effect of ivabradine in randomized clinical trials (e.g. 10.9 bpm in the SHIFT trial) and is comparable in magnitude to the heart rate reduction by ivabradine.”

Comment #3: I'd appreciate if someone with more knowledge than me was able to comment on the plausibility of the bidirectional analyses. Is it plausible to suggest that there could be bidirectional effects of AF on heart failure? Does AF sometimes precede HF, and sometimes HF precedes AF? If not, this doesn't invalidate the analyses (could be that the genetic variants are picking up an effect of subclinical HF on AF risk or vice versa), but it changes the interpretation - in my (limited) experience, true bidirectional effects are rare.

Author response: Thank you for this remark. In general clinicians are not surprised by this bi-directional relationship as it is consistent with clinical observations. More rigorously, these results are consistent with observations from the Framingham longitudinal study as mentioned in the “Results / Bi-directional MR” section:

“These results are concordant with observational longitudinal studies that have observed an increased incidence of atrial fibrillation in new heart failure patients and vice versa and where both diseases are often diagnosed on the same day 26.”

Furthermore, the shared risk factors and etiology of AF and HF is well established in the literature. For a recent review, see:

Carlisle, Matthew A., Marat Fudim, Adam D. DeVore, and Jonathan P. Piccini. 2019. “Heart Failure and Atrial Fibrillation, Like Fire and Fury.” JACC. Heart Failure 7 (6): 447–56.

Comment #4: This is really picky, but generally the term "2SLS method" implies a continuous outcome and a linear regression model. If you used linear regression and treated the binary outcome as continuous, then fair enough (it doesn't make much difference). But if you used logistic/Cox PH regression, then "two-stage method" is my preferred term (others have been suggested).

Author response: We agree with this comment and the importance of avoiding misleading wording of statistical methods. We have adapted the text to use two-stage method when the 2nd stage involves a logistic regression. Concretely, the text in “Methods / Mendelian randomization” now reads:

“For the heart rate Genetic Risk Score (GRS), we used the two-stage method with individual level data and the effect estimates are for a 11.1 bpm reduction in heart rate as well”

Otherwise, I don't have much else to say - it was an interesting read!

---

Stephen Burgess

Reviewer #3: 

The authors present their findings on using genetic variation in HCN4, the gene which encodes the target of ivabradine, to replicate the findings of previously published drug trials of ivabradine. Major strengths are the various complementary analyses using large-scale data sources and the detailed documentation. Overall the manuscript is well-written. Some observations and questions:

Author response: We thank the reviewer for the comments on the manuscript.

Comment #1: It seems that the UK Biobank was a large contributor to many of the sources of summary statistics. It would be good to provide the reader insight in the % sample overlap (always with respect to the larger study) for the various data sources which are combined across the different analyses.

Author response: We now provide in Supplementary Table 7 an estimate on the overlap of UK Biobank samples from the large consortiums based on the published numbers of cases and controls.

The “Supplementary Methods / External summary statistics” section now reads:

“To provide insight into the overlap between samples from the external summary statistics and the UK Biobank, we summarized the shared number of cases and controls in Supplementary Table 7.”

And the table shows an overlap in cases ranging from 13.7% (HERMES) to 24.5% (Roselli et al.) and an overlap in controls ranging from 39.3% (Nielsen et al.) to 64.1% (Roselli et al.).

Comment #2: Given that the genetic risk score for heart rate was derived from a GWAS meta-analysis where the UK Biobank formed the discovery stage, any MR analyses performed in the one-sample setting of the UK Biobank with this score might suffer from the Winner’s Curse. How would this influence the reported results?

Author response: We thank the reviewer for highlighting this important consideration. In our analyses based on the heart rate score, we have used 5,000 bootstrap resamples to empirically estimate the variance of the causal estimate. This procedure is briefly described in the “Supplementary Methods / Mendelian randomization” section. We believe that this procedure accounts for the possible inflation of the weights used to build the GRS. Furthermore, our MR analyses of the effect of heart rate is consistent with other methods that have been estimated using individual variant summary statistics (Supplementary Table 6).

Comment #3: On page 10 the authors describe how adjusting for atrial fibrillation would be problematic if both the SNP and heart failure collide on AF, which would introduce collider bias. However, isn’t it equally likely and problematic that, if the SNP has an effect on AF, and AF has an effect on heart failure, that both the SNP and the confounders of the AF-heart failure association would collide on AF (leading to collider bias when you adjust for AF)?

Author response: We believe that the reviewer’s assessment that adjusting for AF in estimating the SNP-HF relationship would be susceptible to collider bias.

In the manuscript, we propose the cause-specific competing risk model of incident HF as a means of estimating the effect of the SNP without the mediation by AF and without problematic statistical adjustment (Table 1). This result from the competing risk model is also supported by the mtCOJO analysis reported by the HERMES case-control consortium as mentioned in the “Results / Genetically predicted effect on efficacy endpoints” section (see ref. 14). We believe that the evidence from this method that is robust to collider bias as well as our results from the competing risk model reduce the risk of effect distortion due to collider bias.

Comment #4: The interpretation of main effect estimates are less straightforward when interaction effects have been added to the model. Therefore, please be explicit how the reader should interpret the sentence providing both the main and interaction (with AF) estimates between rs8038766 and heart failure.

Author response: We thank the reviewer for this comment which helped present the results of this analysis more clearly. We have adapted the text to report the raw beta coefficients as well as the OR for the SNP in individuals with and without atrial fibrillation. Specifically, the text now reads:

“In a model including the interaction term between rs8038766 and atrial fibrillation, the estimated coefficient for the variant was -0.136, 95% CI -0.205, -0.067, (P=0.00011) and the interaction term coefficient was 0.110 95% CI 0.005, 0.215 (P=0.04). These coefficients correspond to an estimated OR of the SNP on heart failure of 0.87 in individuals without atrial fibrillation and 0.97 in individuals with atrial fibrillation.”

Comment #5: Figure 1: Why was rs7174098 used for just one outcome in MEGASTROKE?

Author response: The selected variant, rs8038766, was not in the MEGASTROKE results file for large artery stroke which we downloaded from http://www.megastroke.org/. It was available for the other outcomes. We selected a very highly correlated variant to be used instead. We added this precision to the manuscript. The Figure legend now reads:

“* rs7174098 (LD r2=1 in 1000 genomes Europeans) was used instead of rs8038766 as the latter was unavailable in the MEGASTROKE summary statistics for this outcome.”

Comment #6: Supposedly you choose the transethnic data of MEGASTROKE for its large number of cases. However, rs8038766 need not necessarily be a strong genetic proxy for HCN4 in non-Europeans. Does using METASTROKE's European dataset show comparable results?

Author response: The European dataset did show similar results and we indeed decided to only present the trans-ethnic results for brevity and given the larger sample size.

Specifically, the European OR for cardioembolic stroke was 1.07 (1.01, 1.12) p = 0.013. 

Comment #7: Please report (perhaps in supplemental material) whether the various GWAS meta-analyses were based on incident or prevalent cases and whether recurrent events were included.

Author response: Large GWAS meta-analyses typically use a combination of prevalent, incident and recurrent events in a case/control association model to increase the number of cases. For example, in the HERMES consortium analyses, 23 of the 51 included studies (45%) were RCTs suggesting that some events were incident and other were recurrent (or “worsening” events) with respect to the inclusion criteria of the underlying studies. Among the remaining studies are 16 (31%) cohort studies which included both prevalent and incident events. To summarize the data we used for large consortia was unspecific with respect to the event type and represented case/control analyses based on liberal case definitions.

We have not updated the manuscript because we feel that referring to the original publication is the best way to gain insight into the individual components of the meta-analysis on a case-by-case basis.

Comment #8: There exist MR methods which can incorporate correlated genetic variants to boost power. Did you consider these methods for HCN4-variants?

Author response: Because HCN4 is a short gene with a large constraint score (pLI = 1 in gnomAD), there are few functional variants that make good genetic instruments. In other words, we believe that there are few variants of functional relevance (refer to Comment #1 from reviewer #2 above). Additionally, Figures 1 and 2 report observational effects for a single variant based on various studies. We believe that for these non-MR estimates using a single variant adds clarity and avoids difficulties that could arise from using weighted scores like variability in variant availability in summary statistics, differences in LD patterns between studies. The other MR analyses we conducted were not “cis-MR” analyses and included independent variants from across the genome.

Comment #9: Table 1 describes the ‘genetic model for SHIFT/SIGNIFY’. Please be explicit this only refers to the outcome definition (and intervention), i.e., not also the inclusion criteria for participants.

Author response: We thank the reviewer for this suggestion that clarifies an important aspect of our manuscript. We have added the following text to the Table 1 legend:

“** Our model aims to match the outcome and exposure of interest from the SHIFT and SIGNIFY trials, but we did not emulate the trials in any other way such as by matching the inclusion / exclusion criteria.”

Comment #10: The selected population of the UK Biobank may give rise to issues like selection bias, also for MR studies. Could this have influenced your results?

Author response: We agree with the reviewer’s comment. In the discussion, the manuscript reads “The MR estimates from the UK Biobank are also based on mostly healthy individuals with a low heart rate (mean of 69 bpm) possibly limiting clinical interpretation 33.” Which aims at highlighting the enrichment of healthy volunteers in the UK Biobank. In the “Study limitations” section, we also mention the possible differences in ethnicity and clinical profile: “We also used data from individuals of predominantly European ancestry both in the UK Biobank and in summary statistics from large GWAS consortia which could limit the generalizability of our results to other populations both in terms of clinical profile and ancestry.”.

We believe that these aspects are to be considered in the interpretation of our findings. Nonetheless, we combined evidence from many independent data sources and analytical models and our results are in line with the observations from ivabradine RCTs. Hence, in this case, the risk of a selection bias that would lead to incorrect conclusions is low.

Minor:

Comment #11: Please mention that the heart rate GRS includes variants which are just relatively independent (r2<0.1). Lower r2 thresholds are now typically advised for MR (e.g., 0.001). In extension, please be more explicit regarding the independence of the various sets of instruments used in the bidirectional MR analyses.

Author response: We thank the reviewer for this comment and have adapted to text to highlight the r2 threshold. The text in “methods > statistical analyses” now reads:

“For the construction of the heart rate Genetic Risk Score (GRS), we used 64 previously reported genome-wide significant heart rate associated SNPs (with r2 < 0.1) 16.”

Additionally, we respectfully disagree with the advice of using LD threshold as low as 0.001. To support our claim, we simulated genotypes for two independent SNPs with MAF 0.2 and corresponding to 1000 individuals. We then repeated this simulation 10,000 times and estimated that the probability of obtaining an r2 less than 0.001 by chance in this setting is around 32%. The R code for this short simulation is as follows:

> set.seed(5)

> n <- 1000

> n.sim <- 10000

> mafs <- c(0.2, 0.2)

> r2 <- sapply(1:n.sim, function(x) cor(rbinom(n, 2, mafs[1]), rbinom(n, 2, mafs[2])) ** 2)

> sum(r2 > 0.1) / length(r2)

[1] 0

> sum(r2 > 0.01) / length(r2)

[1] 0.001

> sum(r2 > 0.001) / length(r2)

[1] 0.3159

In light of this result, we believe that a r2 threshold of 0.001 may be too strict and lead to the exclusion of independent variants.

For the bi-directional analyses, the variants were selected using an r2 threshold of 0.15 as described in the “Supplementary methods > bi-directional MR” section.

Comment #12: Perhaps of interest, FINNGEN could serve as an additional source of publicly available summary statistics for (ICD-code based) heart failure

Author response: We thank the reviewer for this suggestion which may be useful for future work. For the current project, we feel that the HERMES case/control and UK Biobank were sufficient to support our claims.

Comment #13: For the analyses with the GRS it seems two-sample methodology was applied in the one-sample MR setting (e.g., Supplemental Table 5). Perhaps of interest: this (very) recent preprint (https://www.biorxiv.org/content/10.1101/2020.05.07.082206v1) suggests that particularly the MR-Egger method may be easily biased in this setting. Calculating the I2 may give insight.

Author response: We thank the reviewer for pointing out this recent preprint. As the methods other than MR-Egger seem to be less affected by this problem, we feel that our results are unlikely to be strongly distorted as we report estimates based on various MR techniques (IVW, MR-Egger, contamination mixture model, MR-PRESSO) and the 2-stage estimate based on the GRS. As such, we are in favour of keeping the MR-Egger results presented alongside with the other models in Supplemental Table 5. 

Comment #14: Not a fan of ‘non-significant trend’ – perhaps rephrase to ‘provided (very) weak evidence’?

Author response: We agree with the reviewer that this wording has been abused. In this context, we used it because we show, later in the manuscript, that there is indeed a significant association between the HCN4 variant and heart failure when the increased risk in atrial fibrillation is properly accounted for. To avoid the stigma associated with this term, we have reworded as follows:

“We tested for association of the heart rate-reducing allele at the HCN4 variant rs8038766 with combined prevalent and incident heart failure in the UK Biobank and found week evidence of association (OR = 0.96, 95% CI 0.91-1.00, p=0.071) (Figure 2).”

Comment #15: With regard to the kinship threshold of >0.0884 – wouldn’t this reflect 2nd degree relationships or more (rather than ‘or less’)?

Author response: We thank the reviewer for spotting this mistake. The text now reads: “To avoid including related individuals, we used the kinship estimates from the UK Biobank and randomly selected an individual for pairs with a kinship coefficient > 0.0884 consequently keeping only individuals with 2nd degree relationships or less.”.

Comment #16: Please note that quintiles are not the group themselves but rather the cut-offs to define these groups

Author response: We thank the reviewer for this comment. 

The “Methods / Statistical analyses” section now reads: “We split the participants based on the GRS quintiles with the group formed by the 5th GRS quintile (and above) corresponding to the higher heart rate group and the odds ratio for CAD, heart failure and atrial fibrillation were obtained by comparing the first 4 groups individually to the 5th group used as reference in logistic regression.”.

We have also adapted the wording in the Supplementary Figure 2 legend which now reads: “Effect of heart rate genetic risk score groups based on quintiles on atrial fibrillation, heart failure and coronary artery disease in the UK biobank dataset. For every outcome, the highest heart rate group (determined by the 5th quintile) is used as the reference group and the reported odds ratios are adjusted for age, sex and the first 10 principal components.”.

Comment #17: The abbreviation GRS is introduced twice in the methods section

Author response: We thank the reviewer for noticing this. The mention of “GRS” in the “Methods / Mendelian randomization” section now uses the acronym.

---

## [Decision Letter · Decision Letter 1]

1 Jul 2020

A genetic model of ivabradine recapitulates results from randomized clinical trials

PONE-D-20-09954R1

Dear Dr. Dubé,

We’re pleased to inform you that your manuscript has been judged scientifically suitable for publication and will be formally accepted for publication once it meets all outstanding technical requirements.

Kind regards,

Ify Mordi

Academic Editor

PLOS ONE

Reviewers' comments:

Reviewer's Responses to Questions

**Comments to the Author**

1. If the authors have adequately addressed your comments raised in a previous round of review and you feel that this manuscript is now acceptable for publication, you may indicate that here to bypass the “Comments to the Author” section, enter your conflict of interest statement in the “Confidential to Editor” section, and submit your "Accept" recommendation.

Reviewer #1: All comments have been addressed

Reviewer #2: All comments have been addressed

Reviewer #3: All comments have been addressed

2. Is the manuscript technically sound, and do the data support the conclusions?

Reviewer #1: Yes

Reviewer #2: Yes

Reviewer #3: Yes

3. Has the statistical analysis been performed appropriately and rigorously? 

Reviewer #1: Yes

Reviewer #2: Yes

Reviewer #3: Yes

4. Have the authors made all data underlying the findings in their manuscript fully available?

Reviewer #1: No

Reviewer #2: Yes

Reviewer #3: Yes

5. Is the manuscript presented in an intelligible fashion and written in standard English?

Reviewer #1: Yes

Reviewer #2: Yes

Reviewer #3: Yes

6. Review Comments to the Author

Reviewer #1: (No Response)

Reviewer #2: Thanks for responding to the comments in a comprehensive way. I particularly liked the justification of the R^2<0.01 threshold rather than R^2<0.001.

Best wishes, Stephen Burgess

Reviewer #3: (No Response)

7. PLOS authors have the option to publish the peer review history of their article (what does this mean?). If published, this will include your full peer review and any attached files.

Reviewer #1: No

Reviewer #2: **Yes: **Stephen Burgess

Reviewer #3: No

---

## [Editor Report · Acceptance letter]

7 Jul 2020

PONE-D-20-09954R1 

A genetic model of ivabradine recapitulates results from randomized clinical trials 

Dear Dr. Dubé:

I'm pleased to inform you that your manuscript has been deemed suitable for publication in PLOS ONE. Congratulations! Your manuscript is now with our production department. 

Kind regards, 

on behalf of

Dr. Ify Mordi 

Academic Editor

PLOS ONE